# An Event–Link Network Model Based on Representation in P-Space

**DOI:** 10.3390/e27040419

**Published:** 2025-04-12

**Authors:** Wenjun Zhang, Xiangna Chen, Weibing Deng

**Affiliations:** 1School of Medical Information Engineering, Anhui University of Chinese Medicine, Hefei 230012, China; wenjun@ahtcm.edu.cn; 2College of Science, Henan University of Engineering, Zhengzhou 451191, China; 3Key Laboratory of Quark and Lepton Physics (MOE) and Institute of Particle Physics, Central China Normal University, Wuhan 430079, China

**Keywords:** complex network, event–link model, transport network, representation in P-space, growing mechanism

## Abstract

The L-space and P-space are two essential representations for studying complex networks that contain different clusters. Existing network models can successfully generate networks in L-space, but generating networks in P-space poses significant challenges. In this study, we present an empirical analysis of the distribution of the number of a line’s nodes and the properties of the networks generated by these data in P-space. To gain insights into the operational mechanisms of the network of these data, we propose an event–link model that incorporates new nodes and links in P-space based on actual data characteristics using real data from marine and public transportation networks. The entire network consists of a series of events that consist of many nodes, and all nodes in an event are connected in the P-space. We conduct simulation experiments to explore the model’s topological features under different parameter conditions, demonstrating that the simulation outcomes are consistent with the theoretical analysis of the model. This model exhibits small-world characteristics, scale-free behavior, and a high clustering coefficient. The event–link model, with its adjustable parameters, effectively generates networks with stable structures that closely resemble the statistical characteristics of real-world networks that share similar growth mechanisms. Moreover, the network’s growth and evolution can be flexibly adjusted by modifying the model parameters.

## 1. Introduction

Complex networks have emerged as a powerful tool for analysis and modeling across various disciplines [1,2,3,4,5], including sociology [6,7,8], biology [9,10,11,12], computer science [13], transportation [14,15,16], and physics [17,18,19]. Their ability to help researchers better understand and predict the behavior of complex systems has led to the growing adoption of complex network theory in diverse network analyses [20]. The rapid development in this field has resulted in numerous models, theories, and algorithms designed to capture the unique characteristics and structures of various real-world networks [21,22,23,24,25,26].

Understanding the structure of complex systems and their underlying universal laws has long been a focal point of research. By studying network generation mechanisms, we can gain insights into how real networks function. Revealing the internal structure and evolution of real-world networks through models has become a prominent topic in complex network research [27]. These models aim to represent real-world networks abstractly, such as the classical regular network, random network [28], small-world network [29], and scale-free network [30].

Studying how real networks operate through their generation mechanisms presents a valuable research opportunity. Additionally, simulating these networks makes abstract concepts more tangible and relatable. Empirical explorations have highlighted several commonalities in real networks, network degree distributions often follow a power law [3,11,30,31,32], and these networks typically exhibit high clustering coefficients and short average paths. These characteristics have inspired researchers to develop general models to explain them. Among the most noteworthy are the Watts–Strogatz (WS) model and the Barabási–Albert (BA) model. The WS model, developed by Watts and Strogatz, bridges regular and random networks, proving to be more representative of real networks. Examples include food networks, airport networks, and word co-occurrence networks, which all show small-world properties. The BA model, which was proposed by Barabási and Albert, demonstrates that networks adhere to power-law distributions, where growth and preferential attachment serve as two fundamental mechanisms for complex network evolution.

Classical evolutionary models have laid a significant foundation for understanding real network systems, paving the way for studying complex network models. Various optimized evolutionary models derived from these foundational concepts aim to describe real-world networks in a more comprehensive manner [33]. From the perspective of network clustering, Holmes [34] enhanced the BA model by introducing a triangle formation mechanism, known as Triad Formation, into the preferential connection rule. This adaptation resulted in networks with high clustering. Inspired by the BA model, Lu et al. [35] extended the “preferential connection” mechanism to the particle swarm organization process, leading to scale-free network models characterized by high aggregation.

Several node aging models have been proposed when considering preferential attachment within networks. Amaral [36] introduced a model based on the Southern California power grid and actor cooperation networks, where node aging affects connectivity. The research by Dorogovtsev [37,38] indicated that the probability of preferential connection decreases with node age. Additionally, Saramäki [39] developed a local area network model that allows new nodes to preferentially connect to an existing local network, exhibiting scale-free properties. In terms of growth, Kim et al. [40] proposed a complex network evolution model characterized by accelerated node growth, where the size of the growing network expands linearly. Liu et al. [41] discovered that the rate of edge evolution in many real-world networks is significantly greater than the rate of node increase, resulting in a new network growth evolution model. Recognizing that it is impractical for a real network to consistently add a fixed number of nodes during each period of its evolution, Wang et al. [42] proposed a growth model for complex networks that enhances the accelerated increase in the number of nodes. In their model, the number of network nodes grows geometrically, reflecting the evolving nature of real networks. Kharel introduced a new degree-preserving growth model [43] that assumes that the total average degree in the network remains constant, allowing for the generation of various models based on different parameters.

Complex networks are fundamentally connected to graph theory [44,45]. Mathematically, a network is defined as a graph G(V,E) characterized by a set *V* of nodes and a set *E* of edges [44,45]. Depending on the aim of the network analysis, the graph can be represented in different types of representation spaces. The most commonly utilized network representation spaces are L-space and P-space [46,47,48,49]. These two representations are particularly prevalent in transportation networks, including railways [16,50,51,52,53,54,55], ship transport [56], subways [57], and bus networks [15,58,59,60,61,62]. Similarly complex network construction occurs in urban expansion [63] and interpersonal relationships [64]. A transportation network in physical L-space refers to the layout of infrastructure such as roads and rail lines. In this context, nodes represent fixed locations, like intersections and stations, while edges denote direct physical connections between these locations. In contrast, a transportation network from a passenger-centric perspective P-space focuses on the travel experience. Here, nodes correspond to the origins and destinations of trips, and edges represent direct travel paths, such as bus routes or subway lines, which are designed to optimize service efficiency and accessibility. In networks modeled in P-space, stops are treated as nodes, and edges connect every pair of stops that are part of the same route, resulting in a complete subgraph. In L-space, however, stops are connected only to those nearby along the route. This representation effectively transforms the distances and positions between nodes into a cohesive network structure, facilitating a deeper analysis of network topology and dynamic behavior. It serves as a valuable tool for studying and exploring real-world networks.

The P-space representation is similar to a hypergraph, which contains a series of hyperedges. Each hyperedge contains nodes that are connected by this hyperedge [65,66]. Hypergraphs emphasize multimode in different nodes and multidimensional relation between this nodes, using hyperdegree to indicate the number of lines connected to a particular node. In contrast, networks represented in P-space focus on single-mode topological analysis, where the degree reflects the number of ports connected to a given port in the context of studying transportation networks. Zhang [67] presented a growth model grounded in hypergraph theory that incorporates a double-priority attachment mechanism. Moreover, Wang [68] proposed a hypernetwork model. They set a fixed number *m* of notes nodes connected to one existing node by hyperedges; the existing node is chosen proportional to its hyperdegree. Guo [69] improved this model by changing the fixed number of nodes at each step to a non-uniform distribution. They defined the positive integers $\eta_{N(t)}$ and $\epsilon_{N(t)}$ to represent the number of new nodes and existing nodes, respectively. Chen Avin [70] proposed a conceptual framework characterized by an approximate model while positing that not every step necessitates the inclusion of new nodes within the network. Rather, there are instances in which existing nodes may interact to establish a novel hyperedge, thereby enhancing the network’s structural complexity without the continual addition of new components.

Most existing models rely on two main mechanisms: growth and preferential connection. It is clear that the processes used to add nodes in real networks are not fixed but follow some specific distributions, and the connections formed by edges can also vary. Therefore, assuming that real networks increase by a constant number of nodes during each period or that nodes are added uniformly throughout the evolution process is unrealistic. We aim to develop a more accurate network model based on real-world data by considering the simultaneous growth of both edges and nodes and integrating these insights with established high-aggregation network growth models.

In this study, we focus on the representation of these networks in P-space. We propose a P-space-based model capable of replicating the characteristics of accurate data and generating a network that resembles an actual system. In the P-space representation, all nodes on a single line are interconnected within a complex network (shown in Figure 1), this line can be regarded as the event that links all nodes, and a series of events can form a network. To simulate the network generation mechanism, we utilized six datasets, one of which is the worldwide marine transportation network (WMTN), while the other five are public transport networks (PTNs). The new model network aligns more closely with the growth mechanisms observed in real networks, demonstrating a high degree of similarity and facilitating further exploration of the real world.

The rest of this article is organized as follows. In Section 2, we introduce the data and model. In particular, some characteristics of transport data are presented in Section 2.1, and the construction of the model is described in Section 2.2. In Section 3, we provide some topological properties of networks and models. Finally, the last section is devoted to the conclusions.

## 2. Data and Model

### 2.1. Data

The shipping data used in this study were collected from 42 prominent maritime shipping companies, such as COSCO, ANL, APL, HNM, etc., comprising 632 seaports and 2283 shipping lines (routes). We studied the properties of these data in L-space in [14]. However, P-space is more concerned with the change in routes rather than the conversion between ports. The public transport data were collected from bus routes from China’s five largest cities. Compared to air and metro, marine and bus routes contain a more diverse number of nodes. Information on the data is shown in Table 1. In this study, we use nodes to represent stations and seaports.

**Data processing format:** In this study, we present information gathered regarding lines that connect seaports or bus stations. Each line is made up of ports or bus stations, and each is assigned a unique identifier. A line may traverse the same port multiple times, indicating that the line’s length is equal to or greater than the number of ports it encompasses. It is important to note that ports may have different names across various companies, which should be considered when identifying them.

To quantitatively analyze the transport data, we utilize complex networks to visually represent both types. In this representation, edges signify stations or ports involved in the same line, while nodes represent stations or ports that are located on a line. The process of transforming a line into a network in P-space is shown in Figure 1. By mapping this information, we can obtain the largest giant component of graph representing these nodes. Then, we generate a worldwide marine transport network and public transport network. The properties of these networks will be discussed further in Section 3.

**Table 1 entropy-27-00419-t001:** The number of lines $N_{l}$ and nodes *N*, average line length 〈L〉, maximum line length of a marine line $L_{max}$, and average probability 〈p〉 of a new node in the lines of the worldwide marine transportation lines and public transport lines of China’s five largest cities.

Properties	Marine	Beijing	Shanghai	Guangzhou	Shenzhen	Chongqing
$N_{l}$	2284	1817	1449	1522	1036	915
*N*	632	12,579	13,133	8416	5691	4505
〈L〉	6.98	25.77	21.75	23.84	30.25	17.19
$L_{max}$	22	95	85	74	102	60
〈p〉	0.0395	0.268	0.416	0.232	0.182	0.286

### 2.2. Model

**Empirical exploration:** Understanding the fundamental properties of the data is essential for developing a model that accurately represents real-world networks. With this knowledge, we can construct a network evolution model based on the data. For instance, we need to examine the distribution of participating nodes within the dataset or determine the relationship between the numbers of new and existing nodes involved in each event. This approach will enable us to create a coherent and consistent network. Consequently, conducting some empirical exploration is necessary. Figure 2 shows the distribution of the number of ports and stations on a single line, respectively. In both cases, the frequency is notably higher near some fixed-value stations or ports in one line, while it tends to be lower for lines that contain a large number of ports or stations. Moreover, the distribution of the ports or stations in one line within this interval can be effectively modeled with a normal distribution represented by the equation $p\propto e^{-{\left(x-\mu \right)}^{2}/\left(2\sigma^{2}\right)}$. The fitting parameters μ and σ represent the mean and standard deviation. Both values of marine and public transport lines are shown in Figure 2. These two parameters are essential to the network construction process. However, it should be noted that the parameter μ is not the average number of the line nodes. Because the range of both sides of μ in the probability of the number of line’s nodes is not symmetrical, the left range is from 2 to the peak, while the right range is from the peak to $n_{max}$. The minimum number of nodes on the line is 2, but the maximum number of nodes on the line depends on the actual need for data simulation, which is defined as $n_{max}$. Furthermore, not all data follow a normal distribution with respect to the number of nodes. Some data, such as computer science collaboration networks, follow a power-law distribution [64], while the number of participants in historical events conforms to an exponential distribution [71].

Upon examining the data, it becomes clear that the number of nodes is less than the average number of stations multiplied by the number of lines. In the context of real-world social development, we can observe that both familiar and novel nodes often accompany the emergence of new events over time. Therefore, when constructing a network in a structured manner, it is essential to consider both old and new nodes within a single line. In this analogy, a line represents an event, while stations (seaports) correspond to the nodes involved. This approach allows us to evaluate whether the progression of these events is coherent by analyzing the proportion of new nodes introduced within each event. By connecting all nodes from a single event, we create a complex network derived from a series of events. This type of network bears resemblance to those constructed in P-space. We propose that the probability of the proportion of new nodes is denoted as *p*, and we define *p* as follows:(1)$$p\left(i\right)=\frac{nn_{i}}{n_{i}}.$$
where $n_{i}$ represents the total number of nodes (seaports or stations) in the *i*-th event (line), and $nn_{i}$ represents the number of new nodes involved in the *i*-th event. Figure 3 shows the distribution of the proportion *p* of new nodes in different lines, and most of these values are equal to 0. We can use the total number of stations (seaports) after removing duplicates, divided by the value of the total number of stations (seaport) in lines, to calculate the average probability of new points joining the network.(2)$$\langle p\rangle =\frac{N}{\sum_{i}n_{i}}$$
where *N* represents the total number of stations (seaports). The results shown in Table 1 are mostly minor. Due to the limited availability of suitable stations and seaports, many nodes selected for connection are already linked by other lines.

**Model construction:** Based on the above analysis, we propose a simplified data-driven dynamic model, which is referred to as the **event–link model (EL model)**. The evolution mechanism of the network within this dynamic model is as follows: We begin with a simplified initial network and then select a set of nodes to fully connect, incorporating both new and existing nodes. This process resembles an event that includes newly joined and familiar nodes. This method of adding links and nodes to the network aims to replicate the effects observed in our manually obtained statistical data. It is also important to consider the number of nodes selected each time, the number of new nodes introduced, and the choice of existing nodes. The algorithms of the model can be introduced as follows. (1) We define the number of initial nodes $n_{max}$ ($n_{max}$ is determined by the maximum number of nodes in each line), and the initial network *G* is generated by fully connecting the initial nodes.

(2) In the subsequent time step *i*, we generate $n_{i}$ nodes and define the subset *S* as a collection of these $n_{i}$ nodes. **The value of $n_{i}$ is determined by the normalized scale (ratio) of the corresponding distribution depending on real-world circumstances, such as normal or exponential distributions.** For example, we should use the normal distribution when we want to study a transport network and use an exponential distribution when studying event data, as each event contains a different number of nodes. The value of $n_{i}$ falls within the interval $\left(2,n_{max}\right)$, with $n_{max}$ representing the maximum number of nodes and 2 representing the minimum number of nodes in an event. (3) The subset *S* consists of two components: the number of new nodes $nn_{i}$ and the number of existing nodes $no_{i}$, where $nn_{i}$ = *p* × $n_{i}$.

(4) Based on the previous step, we select the number of existing nodes $no_{i}$, which is calculated as $no_{i}=n_{i}-nn_{i}$. **The selection mechanism follows a preferential attachment approach based on the BA model while utilizing the degree distribution.** (5) The new nodes in the set *S* are added to the network *G* and are labeled $n+1,n+2,\ldots ,(n+nn_{i}))$. **Regarding the connection, the nodes in the set *S* are fully interconnected.** (6) Steps (2) through (5) are repeated.

(7) The process ends when the number of nodes in the network *G* exceeds the total number of unique nodes *N* present in the actual data.

Here, the network G can be adjusted by adjusting the five parameters of *n*, *p*, $n_{max}$ and the distribution of the number of nodes in each event that have two parameters, mean μ and standard deviation σ. The process of network generation by the EL model is shown in Figure 4. The main idea of this model is that new nodes join the network by fully connecting to both existing nodes and new nodes. The existing nodes are chosen with a probability proportional to their degree. In this model, we can construct a simulated network G(V,E) corresponding to each parameter derived from the actual data.

During the construction of the network model, at each time step, a selection of nodes will be connected to each other. This selection includes both new and existing nodes. As time progresses, there may be instances where two already connected nodes are selected again, resulting in another connection between them. We refer to this kind of edge as a repeated edge (or repeated link). We define the repeated edge probability $p_{e}\left(t\right)$ for each iteration at each time step in the following manner:(3)$$p_{e}\left(t\right)=\frac{E_{t}^{exist}}{E_{t}}.$$
where *t* indicates the time step, while $E_{t}$ denotes the number of edges formed by the selected nodes at time step *t*, and $E_{t}^{exist}$ represents the number of edges that already exist in network G and reconnect at time step *t*, with the condition $E_{t}^{exist}\leq E_{t}$.

As shown in Figure 5, a simulation of repeated edge probability was conducted, and we find that the repeated edge probability $p_{e}\left(t\right)$ satisfies a log function of the following form: $p_{e}\left(t\right)=0.379-0.0419\log\left(t\right)$. After several simulations, we find that the probabilities follow the formula $p_{e}\left(t\right)=a-b\log\left(t\right)$, where *a* and *b* depend on the parameters in the EL model, and *t* represents the time step. Generally speaking, higher parameters of the average probability *p* of new points lead to smaller values of a and b. The probability of edge duplications generated within a network decreases rapidly with each subsequent step in the initial phase. This is because the network’s density tends to decline with growth if the average degree remains relatively stable. So, we can estimate that the edge number *E* is approximately equal to the average number of edges added in each step multiplied by the total number of steps in this model. The average number of edges is represented by the normalization of the total expression $\frac{x(x-1)}{x}e^{-\frac{-{\left(x-\mu \right)}^{2}}{2\sigma^{2}}}$ evaluated from x=2 to $x=n_{max}$. The number of steps corresponds to the parameter *N* in the EL model, divided by the average number of nodes added in each step. Additionally, the edge count *E* should equal to(4)$$\begin{array}{ccc} E & = & \frac{\sum_{x=2}^{n_{max}}\frac{x(x-1)}{2}e^{-{\left(x-\mu \right)}^{2}/2\sigma^{2}}}{\sum_{x=2}^{n_{max}}e^{-{\left(x-\mu \right)}^{2}/2\sigma^{2}}}\frac{N}{p\times \frac{\sum_{x=2}^{n_{max}}xe^{-{\left(x-\mu \right)}^{2}/2\sigma^{2}}}{\sum_{x=2}^{n_{max}}e^{-{\left(x-\mu \right)}^{2}/2\sigma^{2}}}}\\ & = & \frac{N\sum_{x=2}^{n_{max}}\frac{x(x-1)}{2}e^{-{\left(x-\mu \right)}^{2}/2\sigma^{2}}}{p\sum_{x=2}^{n_{max}}xe^{-{\left(x-\mu \right)}^{2}/2\sigma^{2}}} \end{array}$$
if the network is sparse and the number of events has a normal distribution. μ and σ represent the mean and standard deviation of the normal distribution. If the line of actual data has a high repetition rate, we can adjust the probability *p* in the EL model to obtain an appropriate number of edges.

## 3. Simulation Results and Analysis

### 3.1. Comparison with Other Models

This study introduces several classical models for comparison to analyze the advantages and disadvantages of the EL model more objectively and accurately. These models include real networks, the Erdős–Rényi (ER) model, the configuration model, a small-world network (based on the WS model), a network generated by the BA model, and an EL model.

The WMTN and PTNs represent co-occurrence networks generated from data of a marine line and the public transport lines of China’s five prominent cities. Notably, the original network from the data is not fully connected and contains several small networks comprising two or three nodes, which have a minimal impact on the overall structure. Therefore, we have chosen to analyze the giant connected components.

For the random network (ER model), we maintain the same number of nodes and links as in the actual network, with a fixed connection probability between any two nodes.

The configuration model can be categorized into random networks of different orders based on varying constraints during random scrambling. In this study, we utilize the one-order configuration model, which is a simulated network with the same number of nodes *N* and degree distribution P(k) as the original network. The random scrambling algorithm randomly selects two edges $e_{a,b}$ and $e_{m,n}$ in the network. If there are only two links between the nodes $v_{a}$, $v_{b}$, $v_{m}$, and $v_{n}$, in the next step, we delete $e_{a,b}$ and $e_{m,n}$ and create links $e_{a,m}$ and $e_{b,n}$, so that the configuration model is more consistent with the original network without changing the degree distribution. This algorithm stabilizes each index more quickly with a few scrambles and reduces the computational complexity. Based on calculations and references from previous studies [72], the number of successful scrambles in the one-order configuration model is set to the total number of edges *E* in the real networks.

We generate a graph using the WS model by considering the “six degrees of separation” theory. First, a lattice is created with a specified number of nodes *N* and a defined number of neighbors. The edges of this lattice are then rewired randomly with a probability *p*. In this study, we set *N* equal to the number of nodes in the actual network, with appropriate nearest neighbors and a reconnection probability *p* of 0.2, representing an ideal small-world network model that closely resembles a regular network.

The scale-free network generated by the BA model highlights that many real-world networks exhibit degree distributions that follow a power-law distribution. In this model, every node has connections based on a preferential attachment mechanism, meaning that nodes with higher connectivity are more likely to receive new edges. In this configuration, the total number *N* of nodes in this network corresponds to the number of nodes found in the real-world counterparts. This structure allows for the emergence of hubs, which play a crucial role in the robustness and efficiency of the network.

The last model is the event–link model. According to the algorithm introduced in the previous section, a dynamic network with the same distribution of node numbers as marine lines and public transport lines was simulated. It is important to note that the number of nodes chosen for the EL model corresponds to the number of nodes derived from actual data. Since the number of nodes is selected according to a normal distribution, the total number of nodes in the model will be in the range [N,N+x], where *N* is the total number of nodes in the data and *x* represents a variable number of additional nodes selected each time. This minor adjustment is not expected to significantly affect the overall index results of the modeled network. The results of each model can be found in Table 3 and Table 4.

### 3.2. Topological Properties

We select some common network topologies to compare the functions of different network models that contain the clustering coefficient *C*, the diameter of the network *D*, the average shortest-path distance 〈d〉, assortativity and disassortativity *r*, and the average trapping time 〈T〉. The definitions and formulas are shown in Table 2. Table 3 and Table 4 show the topological properties of the WMTN, the Beijing PTN, and the networks generated by the ER model, configuration model, WS model, and EL model, with similar values of *N* and *E*. *N* is the number of nodes, and *E* is the number of edges. E=〈k〉N/2, where 〈k〉 is the average degree of all nodes.

**Table 2 entropy-27-00419-t002:** The definitions and formulas of the major topological properties of networks. $k_{i}$ is the degree of node *i*, $G_{▵}$ is the number of triangles in the network, $d_{ij}$ is the shortest path length of nodes *i* and *j*, $j_{i}$ and $k_{i}$ represent the degree value of the two nodes corresponding to edge *i*, *M* represents the total number of edges, $s=\sum_{i=1}^{N}\sum_{j=1}^{N}w_{ij}$, $w_{ij}$ is the weight of nodes *i* and *j* (with a value equal to 1 in this study), and $\lambda_{k}$ is the *k*-th eigenvalue of the Laplacian matrix of the network.

Topological Quantities	*N*	Formula
*C*	Clustering coefficient	Ci=3G▵∑i=1nki2
*D*	Diameter of the network	$D=max\left(\{d_{ij}\}\right)$
〈d〉	Average shortest-path distance	$\langle d\rangle =\frac{1}{N(N-1)}\sum_{i\neq j}d_{ij}$
*r*	Assortativity and disassortativity	$r=\frac{M^{-1}\sum_{i}j_{i}k_{i}-{\left[M^{-1}\sum_{i}\frac{1}{2}\left(j_{i}+k_{j}\right)\right]}^{2}}{M^{-1}\sum_{i}\frac{1}{2}\left({j_{i}}^{2}+k_{j}^{2}\right)-{\left[M^{-1}\sum_{i}\frac{1}{2}\left(j_{i}+k_{j}\right)\right]}^{2}}$
〈T〉	Average trapping time	$\langle T\rangle =\frac{s}{N-1}\sum_{k=2}^{N}\frac{1}{\lambda_{k}}$

*C* is the clustering coefficient, which indicates the degree of cohesion among people in a network. In Table 3 and Table 4, the clustering coefficients *C* of the WMTN and Beijing PTN are close to 0.7; combined, they have a relatively small average shortest-path distance, indicating that the WMTN and PTNs have small-world characteristics. Comparing the WMTN and Beijing PTN with the networks simulated with models, it can be shown that the clustering coefficient value *C* in the EL network is close to that of the actual data and is larger than that in the other models. Combined with the analysis of the 〈d〉 variables, this indicates that EL model’s network has small-world characteristics and is more suitable for generating actual networks in P-space. The Watts–Strogatz (WS) model is designed to capture the small-world properties of networks. However, to achieve a high mean clustering coefficient (0.71) using the WS model, the reconnection probability must be reduced to 0.001. This value is not reasonable, so the WS model fails to produce network characteristics that closely resemble those of the actual data.

In the WMTN, the network diameter *D* is 5, which is greater than the value for three of the simulated networks. However, the EL and configuration models have the same diameter (*D* = 5) as that of the WMTN. The Beijing PTN’s diameter is greater than those of all of the simulated networks, but the diameters of the EL model and configuration model are closest to that of the Beijing PTN. The average shortest-path distance can represent the tightness of a network’s connections. Regarding 〈d〉, there is little difference between each model and the real network, and the values of the actual network are slightly greater than those of the simulated networks. Generally, a small-world network has a shorter length 〈d〉 and a larger clustering coefficient *C*. All of these models can generate a network with some small-world properties. However, the EL model can generate a network with all small-world properties.

The variable *r* denotes the Pearson correlation coefficient (−1<r<1) of the degree of two nodes that are connected in a network, and it is used to represent the heterogeneity of the network (assortativity and dissortativity) [11,73,74,75]. As *r* approaches 1, it suggests that the network exhibits an associative structure, where nodes of higher degrees are more likely to be connected to other nodes of high degrees. Conversely, if *r* tends toward −1, this indicates disassortativity, meaning that nodes with higher degrees tend to connect with nodes of lower degrees. The *r* values of all networks are close to 0, meaning that they have no obvious assortativity or dissortativity properties.

Among the topological properties, the average trapping time (ATT) 〈T〉 is the average first-pass time from any node random walk to another node in the network, representing the connectivity of the network [76,77]. The 〈T〉 values of the actual network and models were calculated. Both the EL and null networks had values similar to those of the actual data and larger than those of the other networks.

**Table 3 entropy-27-00419-t003:** Topological properties of the WMTN and networks generated by the ER, configuration, WS, BA, and EL models with similar node and edge numbers. *N* is the number of nodes, 〈k〉 is the average degree, *C* is the clustering coefficient, *D* is the diameter of the network, 〈d〉 is the average shortest-path distance, *r* is the assortativity, and 〈T〉 is the average trapping time. The bold values indicate that the EL model can generate a network more similar to the data.

Topological Quantities	*N*	〈k〉	*C*	*D*	〈d〉	*r*	〈T〉
WMTN	632	33.72	**0.7099**	5	2.426	−0.1201	**1357**
ER	632	33.72	0.0538	3	2.102	0.0002	334.4
Configuration	632	33.72	0.3510	5	2.239	−0.1925	**1275**
WS	632	34	0.3902	3	2.392	−0.0041	338
BA	632	33.52	0.1324	3	2.113	−0.0.017	427
EL	632	34.62	**0.7399**	5	2.279	−0.1396	**1326**

**Table 4 entropy-27-00419-t004:** Topological properties of the Beijing bus network and the networks generated by the ER, configuration, WS, BA, and EL models with similar node and edge numbers. *N* is the number of nodes, 〈k〉 is average degree, *C* is the clustering coefficient, *D* is diameter of the network, 〈d〉 is the average shortest-path distance, *r* is the assortativity, and 〈T〉 is the average trapping time. The bold values indicate that the EL model can generate a network more similar to the data.

Topological Quantities	*N*	〈k〉	*C*	*D*	〈d〉	*r*	〈T〉
Beijing PTB	12,557	87.31	**0.7488**	7	3.606	0.06447	**17,628**
ER	12,557	87.31	0.00692	3	2.534	0.00271	6424
Configuration	12,557	87.31	0.02939	5	2.524	−0.02007	**16,300**
WS	12,557	88	0.3832	4	2.782	0.000306	6467
BA	12,557	87.84	0.0291	3	2.422	0.00197	8475
EL	12,565	87.03	**0.7624**	5	2.521	−0.01056	**19,737**

A comparison of the actual data and the results of the EL models of the other four PTNs is shown in Table 5. All of the simulated networks have properties similar to those of actual data. **In short, the WMTN and PTNs exhibit small-world characteristics; the EL model can simulate similar properties, and its various topological quantities are close to those of the actual data.**

**Degree distribution of the simulation:** To further evaluate the compatibility of EL model with actual networks, we present the degree distributions for all six networks (WMTN and five PTNs) along with their parameters within the EL models, as illustrated in Figure 2. The node degree, denoted as *k*, signifies the number of edges connected to each node, which directly indicates the number of stations (or seaports) that can be reached from a given station (or seaport) without changing buses (or ships). The P(k) represents the probability that a node has a degree of *k*. As shown in Figure 6, both the WMTN and PTNs demonstrate scale-free characteristics, exhibiting power-law degree distributions in medium-to-large degree ranges that affirm the universality of scale-free networks. Notably, this distribution diverges slightly from a traditional power-law distribution, instead reflecting a positive exponential power-rate distribution to the left of its peak probability point. This phenomenon arises when generating complex networks from real data, where a set of nodes are fully interconnected. The distribution of the line lengths indicates that there are relatively few very short lines, leading to a small number of nodes with a minimal degree. A closer examination of the red scatter points in Figure 6 reveals that the EL model portrays a long-tail degree distribution. This overall degree distribution aligns well with the actual network, reinforcing the credibility and plausibility of the EL model. The peak degree probability directly correlates with the distribution of the number of stations along the route, and the EL model successfully replicates a similar distribution to that of the actual data.

## 4. Conclusions

In this study, we address the issue of the normal distribution of numbers of nodes and edges in current network models based on actual data. We propose a novel network model called the EL model, which permits the number of nodes added to the network at each step to follow a normal or exponential distribution, among others. By detailing the construction process of this network model, we demonstrate its enhanced suitability for generating and connecting real-world networks.

Through simulations of the WS network, ER network, BA network, and configuration model, we compare the topological properties of these models with those of the EL network. Our findings indicate that the degree distribution of the nodes in the new model exhibits a power-law characteristic, while the network features a higher clustering coefficient *C* and a shorter average path length 〈d〉, aligning more closely with the characteristics of actual networks. Additionally, the model can generate a network with a similar degree distribution that aligns with the actual network in P-space. The simulation results emphasize the model’s strong applicability and authenticity. Our new model possesses both small-world and scale-free properties, making it more appropriate for real-world networks than the BA and WS models.

Additionally, by adjusting the model parameters *N*, *p*, μ, σ, and $n_{max}$, we can construct relevant real-world networks, enabling the modeling of various complex systems. In our future research, we aim to explore whether this network can more effectively address various real-world network problems, such as communities on the mesoscopic level [78] and robustness in food web networks [79,80].

## Figures and Tables

**Figure 1 entropy-27-00419-f001:**
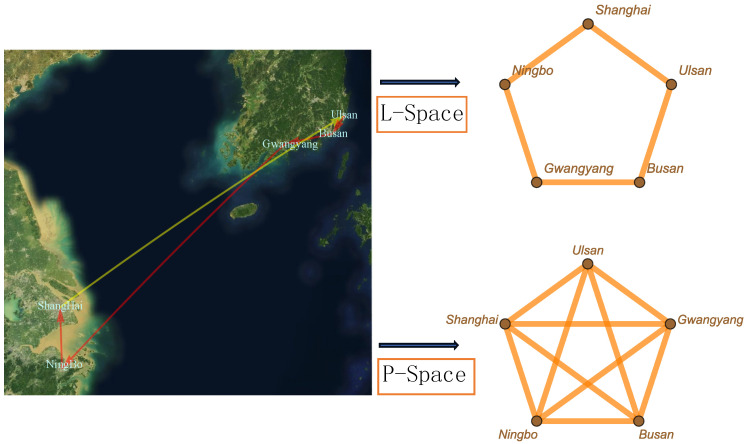
An example of one shipping line converted in to graph in L-space and P-space. The line goes from Ulsan, through the other four seaports, and back to Ulsan. The corresponding network contains five nodes and five edges within L-space, while P-space consists of 10 edges.

**Figure 2 entropy-27-00419-f002:**
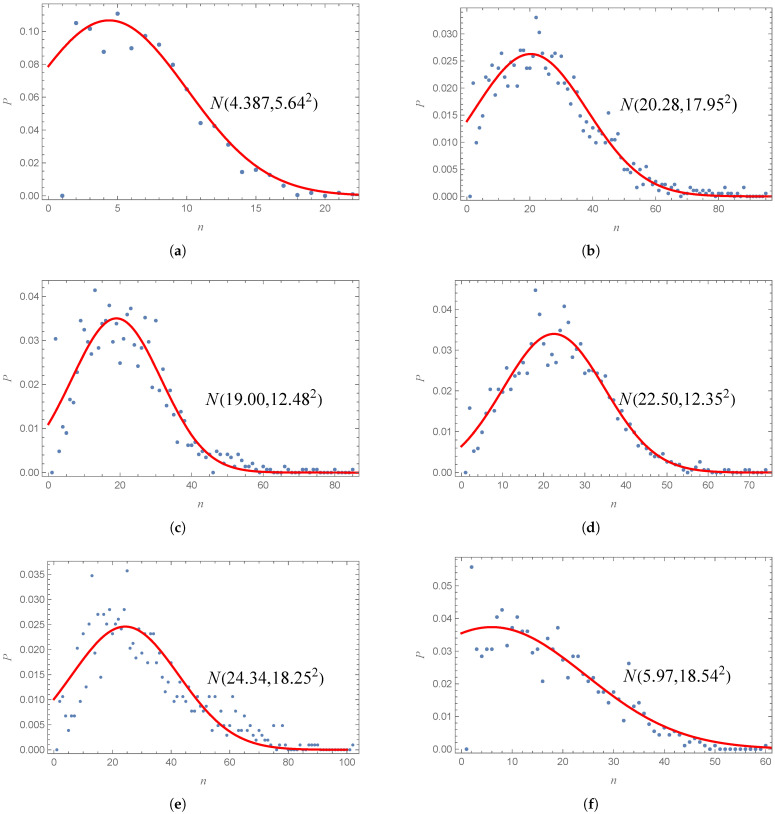
The distribution of the number of stations in each line of the worldwide marine transportation lines (**a**) and bus lines of Beijing (**b**), Shanghai (**c**), Guangzhou (**d**), Shenzhen (**e**), and Chongqing (**f**). The variable *n* represents the number of nodes in a single line, while *p* indicates the probability of this kind of line among all lines. The red lines represent the normal distribution function, with mean μ and standard deviation σ shown as $N(\mu ,\sigma^{2})$ in the corresponding image.

**Figure 3 entropy-27-00419-f003:**
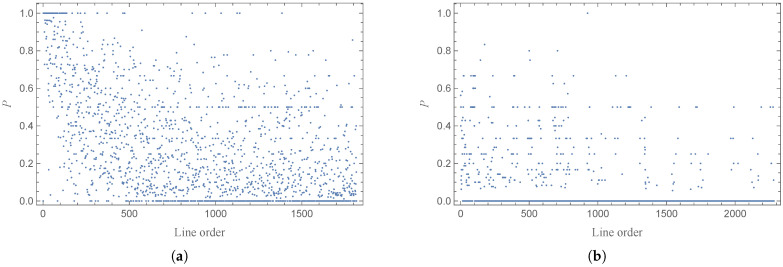
The distribution of the proportion of new nodes in different lines of the marine lines (**a**) and Beijing bus lines (**b**).

**Figure 4 entropy-27-00419-f004:**
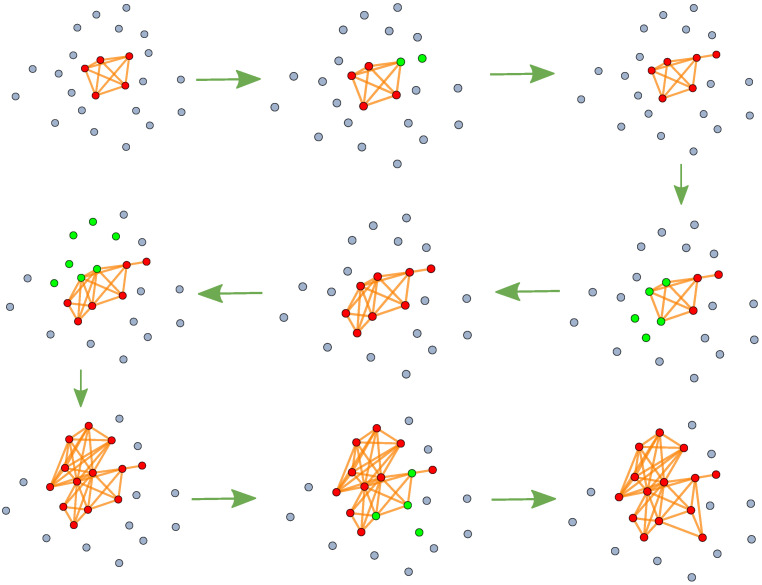
EL models are used to generate flow diagrams of complex networks. the green arrows represent the process of network evolution. The red nodes represent an existing network, the gray nodes are not included in the network, and the green nodes, which were chosen in this step, contain the joining and existing nodes. Finally, a complex network with 14 nodes and 41 edges is generated. The probability of an existing node being selected is proportional to its degree.

**Figure 5 entropy-27-00419-f005:**
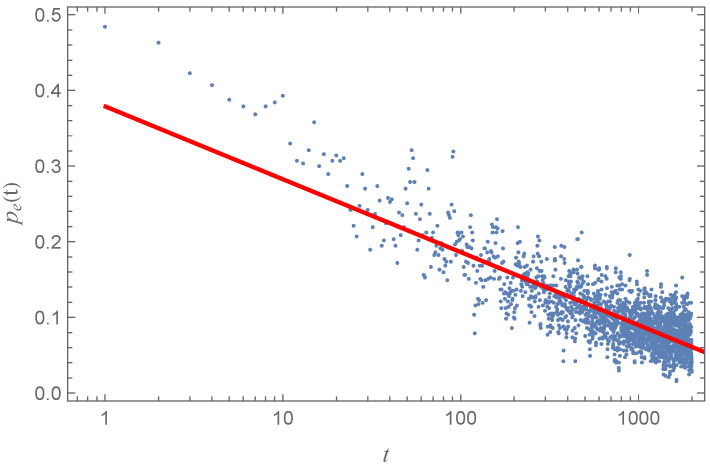
Repeated link probability $p_{e}\left(t\right)$ changed with step *t* in the EL model with parameters of *N* = 10,000, p=0.25, and 20 connected nodes per step. The red line represents the fitted result, which follows the equation $p_{e}\left(t\right)=0.379-0.0419\log\left(t\right)$.

**Figure 6 entropy-27-00419-f006:**
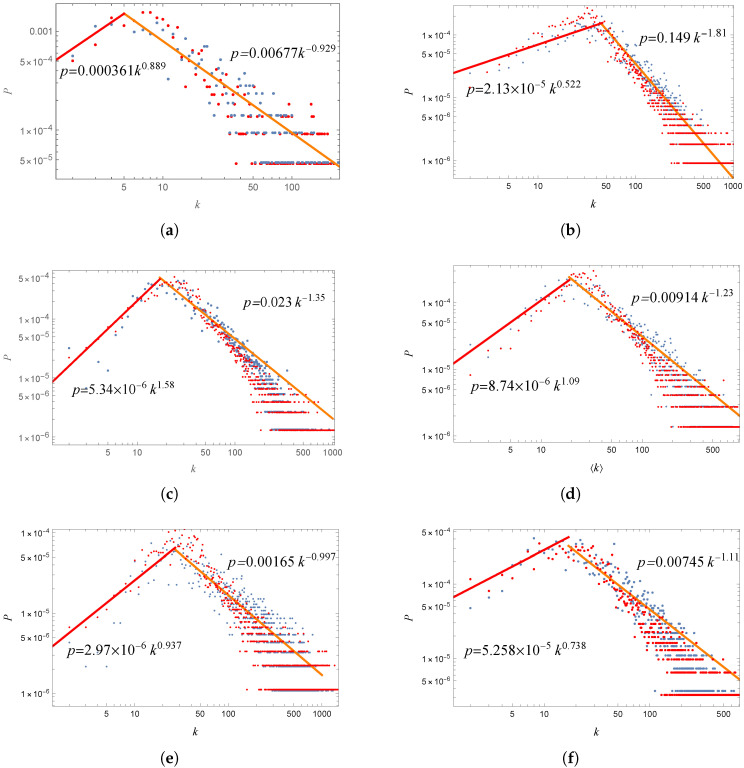
Degree distribution in the marine lines (**a**) and the bus lines of Beijing (**b**), Shanghai (**c**), Guangzhou (**d**), Shenzhen (**e**), and Chongqing (**f**). The inset shows the results on a log–log scale. The blue dots are statistical results from actual networks, and the red dots are the results of networks simulated by the EL model. The red lines represent the fitted lines of degree between (2, peak) of the actual data, and the orange lines represent the fitted lines of degree in the range of (peak, largest) of the actual data. The corresponding fit functions are shown in the figure. *P* and *k* are defined in the text. The *p*-values of all six orange lines are $\left(8.38\;\times \;10^{-20},\;4.10\;\times \;10^{-52}\right)$, $\left(1.13\;\times \;10^{-11},\;1.53\;\times \;10^{-204}\right)$, $\left(3.85\;\times \;10^{-22},\;1.90\;\times \;10^{-158}\right)$, $\left(1.08\;\times \;10^{-16},\;2.85\;\times \;10^{-105}\right)$, $\left(6.05\;\times \;10^{-30},\;2.41\;\times \;10^{-229}\right)$, and $\left(4.64\;\times \;10^{-32},\;1.68\;\times \;10^{-222}\right)$. All the values are far less than 0.05, indicating that all the data adhere to a power law distribution within the appropriate range.

**Table 5 entropy-27-00419-t005:** Topological properties of the Shanghai PTN, Guangzhou PTN, Shenzhen PTN, Chongqing PTN, and networks generated by the EL model with similar node and edge numbers. *N* is the number of nodes, 〈k〉 is the average degree, *C* is the clustering coefficient, *D* is the diameter of the network, 〈d〉 is the average shortest-path distance, *r* is the assortativity, and 〈T〉 is the average trapping time.

Topological Quantities	*N*	〈k〉	*C*	*D*	〈d〉	*r*	〈T〉
Shanghai PTN	13,037	57.38	0.7772	11	4.103	0.1599	19,998
EL of Shanghai PTN	13,039	58.82	0.7792	5	2.688	0.01723	16,717
Guangzhou PTN	8363	86.53	0.7409	7	3.470	0.2546	14,477
EL of Guangzhou PTN	8366	87.47	0.7490	4	2.424	−0.00458	12,913
Shenzhen PTN	5671	164.0	0.6363	6	2.748	0.1757	9829
EL of Shenzhen PTN	5693	158.2	0.7378	4	2.180	−0.09621	12,043
Chongqing PTN	4479	60.96	0.7591	8	3.509	−0.223	8450
EL of Chongqing PTN	4511	58.84	0.7740	5	2.479	−0.02537	7354

## Data Availability

The data used in this study can be obtained from https://github.com/zhangwenjun314/Event_link_model (accessed on 7 March 2021).

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
