# Peer review of "An Event–Link Network Model Based on Representation in P-Space"

_entropy, 2025, doi:10.3390/e27040419_

Round 1

Reviewer 1 Report

Comments and Suggestions for Authors

A novel generative complex network model, referred to as the event-link model, is proposed, which can represent various properties of complex networks within space P. Notably, the mean cluster coefficient of complex networks in P-space approaches 1, and the EL model demonstrates similar outcomes. This characteristic is essential for investigating P-space networks and analogous practical networks. Additionally, the model elucidates the developmental process of such complex networks via a microscopic growth mechanism and broadens the theoretical framework of complex network models.

However, there are a few minor issues need to be addressed before recommending for publication:

1. The author discusses the distribution of the number of participants in each event when introducing the EL model, but it lacks citation. Similar growth mechanism has been observed on evolution of urban systems (Nat. Commun., 2017, 8:1841). Specifically, the statement, “Additionally, not all data follow a normal distribution concerning the number of participants. Some data, such as computer science collaboration networks, adhere to a power-law distribution, while the number of participants in historical events aligns with an exponential distribution,” requires supporting references.

2. In the section addressing the repeated edge probability, the author fails to elaborate on how this is influenced by the El model parameters. Additional details in this area would enhance clarity.

3. The results of degree distribution between the actual data and the corresponding EL model are interesting. However, the authors provide insufficient explanation regarding why the positive exponential power-rate distribution is positioned to the left of the vertex in the degree distribution. A more comprehensive explanation is recommended.

Communities are crucial emerging structure at mesoscopic level (Commun. Phys., 2024, 7:170), is the model able to generate community structure? This aspect should be briefly mentioned or as a future work. 

Reviewer 2 Report

Comments and Suggestions for Authors

In this manuscript, the authors introduce a new model to build network in the so-called space P.
They use a modified version of the preferential attachment models. In particular, they add nodes in cliques (representing new transportation lines) that are completely connected.

The article is clear, even pedantic on some very basic complex networks concepts. It requires a grammatical check.

Content-wise, the proposed model is a modification of the old preferential attachment model proposed by Barabasi and Albert. The, new model add nodes in cliques representing transportation lines. As per Table 3, the only measure in which this model overcomes the traditional models is the clustering coefficient.

I'm concerned by this approach, the outcome seems to represents an artifact due to the fact that the authors add nodes in cliques (increasing artificially the clustering coefficient), jointly with the fact that the real network are considered in space P instead of space L.
I'm pretty sure that, if real networks were built in space L, the traditional models would have been more than appropriate to reproduce the clustering coefficient.

There are many issues that should be fixed:

  • The definitions are not always clear: what are "characters" and "individuals", if they refer to nodes, just call them nodes.
  • The use of a normal distribution (Fig 2) seems unfortunate since it is defined for positive and negative values, maybe a Poissonian distribution would better represents the data.
  • On lines 150-153, the authors should cite the relevant papers.
  • Eq. 3 is defined on a node: "E_t denotes the number of edges of the old node..." and "E_t^exist represents the number of edges of the old node that already exist...". How you go from this to the global value? Why don't you just call them degree at time t and degree at time t-1? Why, in Figure 5, the ratio between the degree at time t and t-1 decrease with time instead of approaching 1? I feel that the definition of repeated edge is not clear enough.
  • What the authors name the "null model" is in fact the "Configuration model". Null model is different in different studies, it represents the simple "random structure" to which one should compare. Please, read at least the Wikipedia article.
  • The authors discard "small networks comprising 2 and 3 nodes". Do you mean "we only consider the giant connected component"?
  • On line 285, a high clustering coefficient does not indicate a small-world characteristic. Consider in fact the WS model with p=0. This type of network will have high clustering coefficient, but the diameter will grow linearly with N.
  • In Figure 6 it is not clear what "statistical results" means. A statistical test should be performed to assess that the distribution is in fact scale free (at least in part of the distribution domain).

Minor issues:

  • The average trapping time is referred to as ATT in the text and <T> in table 3.
Comments on the Quality of English Language

The grammar of the manuscript needs to be improved.

Reviewer 3 Report

Comments and Suggestions for Authors

The research “An Event-link network model based on representation in space P” presents an empirical analysis of some transportation networks and proposes a new model based on network growth. I have major problems.

MAJORS

  • The main problem is that it is unclear whether the proposed model is intended to describe the structure of a network in space L or space P. This should be clarified immediately in a clear and explicit manner. Without this clarification, it also becomes difficult for the reader to understand the meaning of the analyses presented throughout the article.
  • The core idea of tracking the introduction of new nodes is valuable but was not clearly articulated.
  • The analogy between transportation networks and social development should be made more explicit and clearer.
  • Mathematical definitions are unclear. The authors should provide a more structured explanation.

OTHERS

  • Add the axis values l and p in the caption of Figure 2.
  • Add the name of the parameter of the normal distribution in the caption of Figure 2.
  • R19-20: The authors can add ecological networks here. I furnish here some refs:

https://nsojournals.onlinelibrary.wiley.com/doi/10.1111/oik.11139

https://www.sciencedirect.com/science/article/pii/S0022519313002014

  • R139 should be: “Figure 2 shows the distribution of the number of ports and stations on a single line, respectively.”.
  • R141 maybe should be: “; it tends to be lower for lines that contain a large number of distant ports or stations.”.
  • R147: This sentence is confusing: “Because the range of both sides of µ in the probability of the length of the line is not symmetrical.”
  • R150-152: Add references to the distribution results cited in this sentence.
  • I do not understand the context of this sentence: “In the context of real-world social development, we can observe that both familiar and novel characters often accompany the emergence of new events over time.” The analogy between social events and transportation networks is not entirely clear.

  • “Fig. 3 show the distribution” must be "Fig. 3 shows the distribution".

  • "which have removed duplication" must be "after removing duplicates".
  • "Due to the limited number of suitable stations and seaports, the nodes selected for connection often already have connections made by other lines." Maybe "Due to the limited availability of suitable stations and seaports, many nodes selected for connection are already linked by other lines."
Comments on the Quality of English Language

Enghlish may be improved.

Round 2

Reviewer 2 Report

Comments and Suggestions for Authors

This revised manuscript comes with notable improvements both in the readability and in the content. Nevertheless, there are few issues that still need to be clarified before considering the publication.

The main issue comes with the context of the space-P analysis.

The authors propose to study transportation networks in the so called space-P as others have done in the literature (see this manuscript's references). In space-P, nodes belonging to one line are treated as interacting all-to-all, in a sort of collective interaction. This happens to correspond to the definition of a hyper-graph, where a hyper-edge is a set of nodes (as the stations on the same line) that interact at higher level, beyond the pairwise interaction of classical networks (in the current paper called space-L).
See for example the following papers for a review of the topic:
- Ghoshai, Zlatic, Caldarelli, Newman, PRE, 2009
- Battiston, Cencetti, Iacopini, et al., Phys. Rep. 874, 1 (2020).
It seems to happen that two research communities have introduced overlapping definitions with distinct names. This despite the fact that the two research communities work on similar topics.

In the hyper-graph community, researchers have proposed several approaches for a preferential attachment adaptation to this framework. I have not found an approach that replicate the one proposed in this manuscript, but a reference to them with highlighted differences seems important.
See for example:
- Chen Avin; Zvi Lotker; Yinon Nahum; David Peleg IEEE/ACM 2019
- Jian-Wei Wang, Li-Li Rong, Qiu-Hong Deng & Ji-Yong Zhang, Eur. Phys. J. B 77, 493–498 (2010)
- in-Li Guo, Xin-Yun Zhu, Qi Suo & Jeffrey Forrest, Sci Rep 2016

Minor Issues:

- In Line 149 there is a typo: remove `are`.

- In equation 3, $E_t$ refers to the total number of links at a given time-step $t$, while $E_t^\text{exists}$ refers to the **new** edges added at time $t$, that do replicate already existing edges. Given the fact that one quantity describe a cumulative value and the other an increase, I suggest changing the name of the second to $\delta E_t^\text{exists}$.

Non Issues:

In reply to my previous comments, the authors provide a reference where the wording "null model" seems to be used instead of "configuration model". After checking, I can assert that the authors of that reference analyze many null models (as in random models used as benchmark) that are often used in neuroscience, and within these approaches they also describe the "degree-preserving configuration model … often used as the null model". Hence, they also refer to "null models" and "configuration model" as intended by the whole community.

Additionally, the authors reply to another comment: "small-world networks require both high-clustering and short average path distance". This is false since the "small-world" properties only derives from a short average path length (or network diameter) and this can be obtained without increasing the number of triangles. One can consider a regular 2D lattice, this has both high average path length and clustering=0. If we rewire links being careful to connect only distant nodes, we will end up with the small-world effect (due to the increasing number of shortcuts) but maintaining clustering=0 (due to the absence of triangle in the lattice).
Nevertheless, the authors do not mention this in the manuscript, hence no further action is required.

Reviewer 3 Report

Comments and Suggestions for Authors

The difference between networks in space L and P is not clear. Please add two simple and concise sentences explaining what a transportation network in space L is and what a transportation network in space P is.

For example, A transportation network in space L has nodes/bus stops and a direct connection between two bus stops. A transportation network in space P. ....."

Comments on the Quality of English Language

minor revision

Round 3

Reviewer 2 Report

Comments and Suggestions for Authors

In the last review round the authors successfully added reference to the hypergraph community. Personally, I think this bridge enhance the theoretical framework of the manuscript and could potentially attract interests from both worlds.